# Dysphagia Care and Management in Rehabilitation: A National Survey

**DOI:** 10.3390/jcm11195730

**Published:** 2022-09-27

**Authors:** Renée Speyer, Adriana Sandbekkbråten, Ingvild Rosseland, Jennifer L. Moore

**Affiliations:** 1Department Special Needs Education, University of Oslo, 0318 Oslo, Norway; 2Curtin School of Allied Health, Faculty of Health Sciences, Curtin University, Perth, WA 6102, Australia; 3Department of Otorhinolaryngology and Head and Neck Surgery, Leiden University Medical Centre, 2333 ZA Leiden, The Netherlands; 4Aurskog-Høland Municipality, 1940 Bjørkelangen, Viken, Norway; 5Reinforced Interdisciplinary Rehabilitation Aker, 0586 Oslo, Norway; 6Regional Center of Knowledge Translation in Rehabilitation, Aker Hospital, 0586 Oslo, Norway; 7Institute for Knowledge Translation, Carmel, IN 46033, USA

**Keywords:** questionnaire, intervention, screening, assessment, eating and drinking, deglutition disorders, swallowing problems, Norway

## Abstract

Dysphagia care and management may differ between countries and healthcare settings. This study aims to describe the management and care of dysphagia in rehabilitation centres and health houses across Norway. Two national surveys were developed targeting either managers or healthcare professionals. Both surveys focused on staff and client populations; screening and assessment of dysphagia; dysphagia management and interventions; staff training and education; and self-perceived quality of dysphagia care. A total of 71 managers and clinicians from 45 out of 68 identified rehabilitation centres and health houses in Norway completed the surveys. The resulting overall response rate was 72.1%. Significant differences in dysphagia care and management were identified between rehabilitation services across Norway. Rehabilitation centres and health houses often had neither a speech therapist among their staff nor had access to external healthcare professionals. Screening was most frequently performed using non-standardised water swallows and only limited data were available on non-instrumental assessments. None of the respondents reported having access to instrumental assessments. Dysphagia interventions mainly consisted of compensatory strategies, including bolus modification, with very infrequent use of rehabilitative interventions, such as swallow manoeuvres. Although almost half of all respondents perceived the overall quality of care for clients with eating and swallowing problems as good, lack of awareness of dysphagia and its symptoms, consequences and options for treatment may have influenced quality ratings. There is a need to raise awareness of dysphagia and provide training opportunities for healthcare professionals in both screening and assessment, and dysphagia care and management.

## 1. Introduction

Oropharyngeal dysphagia, or swallowing disorders, can result from many underlying diseases (e.g., head and neck cancer, stroke, and neurological diseases) and have a major impact on functional health status and health-related quality of life [1,2]. Dysphagia can result in dehydration, malnutrition, aspiration pneumonia and even death [3,4,5]. Prevalence estimates of dysphagia among the general population vary between 2.3% and 16% [6], whereas prevalence estimates for specific patient populations may be as high as 80% in stroke and Parkinson’s disease patients, and over 90% in patients with community-acquired pneumonia [7]. When estimating dysphagia prevalence for different healthcare settings using meta-analyses, results indicate 36.5% in hospitals, 42.5% in rehabilitation, and 50.2% in nursing homes [8].

Dysphagia management and care pathways may vary considerably between healthcare settings, depending on factors including financial and staffing resources, professional education and cultural differences. Several national and international healthcare surveys have been published targeting management of dysphagia in acute care and hospitals [9,10,11], rehabilitation [12] and nursing homes [13]. Some surveys recruited speech language pathologists from different workplace settings [14] or focused on dysphagia treatment in specific patient populations independent of healthcare setting [15,16].

A recent national survey targeting the management and care pathways for elderly people with dysphagia in nursing homes across Norway identified a need for training and upskilling staff in Norwegian nursing homes and raising awareness of the serious consequences and comorbidities that can result from dysphagia [13]. However, no data have been published on the management and treatment of dysphagia in rehabilitation centres in Norway. Norwegian rehabilitation care is offered in both health houses (‘helsehus’) and rehabilitation centres. Health houses provide interdisciplinary rehabilitation services for municipalities across Norway and provide an intermediate stage between hospital stays and clients’ return to home. The health houses aim to reduce in-hospital stays and rehabilitate clients close to their homes. Rehabilitation centres serve a similar purpose but are often privately owned and form part of Norway’s specialist health services, often focusing on specialised treatment [17,18,19,20,21].

This study aims to describe the management and care of dysphagia in rehabilitation centres and health houses across Norway. Two national surveys were developed targeting either managers or healthcare professionals. The surveys focused on participant characteristics and details on clinical settings, staff and client populations; screening and assessment of eating and swallowing difficulties (dysphagia); management and interventions for people with eating and swallowing difficulties; staff training and education in eating and swallowing difficulties; and self-perceived quality of dysphagia care.

## 2. Methods

### 2.1. Survey Development

Two online surveys were developed based on current literature on dysphagia and expert input from healthcare professionals, rehabilitation managers and academics with expertise in survey development. Nettskjema, a tool for designing and conducting online surveys, was used to administer the surveys. The tool was specifically developed to meet Norwegian privacy requirements and is operated by the University Information Technology Center (USIT) at the University of Oslo, Norway. After piloting both surveys among five content experts, minor revisions were made using provided feedback to improve readability and uniform interpretation of survey questions.

Both surveys included the same four introductory questions about the participant’s workplace (health house or rehabilitation centre), employment position, educational/professional background, and primary responsibilities within the rehabilitation centre (i.e., managerial activities or clinical activities). The last introductory question would lead the participant to either the survey for managers or the survey for healthcare professionals. The final survey for rehabilitation managers consisted of 15 questions about various topics related to the dysphagia care pathway in rehabilitation: rehabilitation service and staffing (5 questions); screening and assessment of eating and swallowing difficulties (6 questions); staff’s training and education in eating and swallowing difficulties (3 questions); and self-perceived quality of dysphagia care (1 question). The survey for healthcare professionals consisted of 22 questions on the following topics: screening and assessment for eating and swallowing difficulties (8 questions); treatment of eating and swallowing difficulties (10 questions); training and education in eating and swallowing difficulties (3 questions); and self-perceived quality of dysphagia care (1 question). The survey for managers included additional questions on rehabilitation service and staffing, while the survey for healthcare professionals included additional questions on screening/assessment and interventions for people with dysphagia. Further details on both surveys can be found in Appendix A.

Where appropriate, the surveys contained short explanations of topics and terminology, such as ‘dysphagia’ or ‘screening and assessment’. The use of professional jargon was avoided as participants were expected to have different educational backgrounds and, consequently, vary in familiarity with medical terms. Therefore, after the introduction of the concept of ‘dysphagia’, the term was replaced by ‘eating and swallowing difficulties’ throughout both surveys.

The survey for managers consisted of 7 multiple choice questions, 5 matrix questions, 2 numeric textbox questions, and 1 ordinal scale question. The survey for healthcare professionals consisted of 12 multiple choice questions, 6 matrix questions, 2 ordinal scale questions, and 2 open-ended questions. Throughout both surveys, participants could elaborate on questions using open comment boxes.

### 2.2. Recruitment of Participants

In December 2021, rehabilitation centres and health houses across Norway were identified. Rehabilitation centres were identified through the public website from the Norwegian Health Network (https://www.helsenorge.no/ accessed on 29 August 2022) and health houses through Google Maps, Google Search Engine, and websites from municipalities across Norway. A total of 32 rehabilitation centres and 90 health houses were found, 36 of which were health houses with rehabilitation facilities. All 68 clinical settings (32 rehabilitation centres and 36 health houses with rehabilitation facilities) were considered eligible for participation and contacted by phone using a standardised protocol and including information about the purpose and content of the study. Centres that could not be contacted on the first attempt were called once more. Two participants were sought from each centre: one staff member mainly involved in managerial responsibilities and one staff member working as a clinical professional. Centres willing to participate provided e-mail addresses from potential participants (with knowledge about management and care of clients with dysphagia) after which additional information was sent, including a link to the online surveys. An information letter was attached with further details on the purpose of the survey, ethical considerations and information about privacy and informant rights. Both surveys were open for respondents from the beginning of February 2022 to mid-March 2022 (six weeks) during which participants received up to three reminders.

### 2.3. Data Analysis

Data were exported from Nettskjema into Excel for data cleaning and quality assessment. Next, data were imported into SPSS (version 28, Chicago, IL, USA). If applicable, participants’ responses in open comment boxes were either recoded into existing response categories or added as a new response option. Descriptive statistical analyses were performed to estimate frequency and percentage distributions. Differences between groups of respondents were investigated using Chi-square test or Fisher’s exact test (if more than 20% of cells in cross-tabulation had expected values less than 5) [22].

## 3. Results

### 3.1. Participants

A total of 55 (80.9%) out of 68 contacted rehabilitation centres and health houses agreed to participate. Three clinical settings failed to provide e-mail addresses. Two clinical settings withdrew as they reported not having clients with eating and swallowing problems among their rehabilitation population, and one setting withdrew due to lack of sufficient knowledge to complete the surveys. The resulting overall response rate was 72.1% (49/68). Clinical settings were spread out over all 11 Norwegian counties. The number of settings per county ranged between 1 and 15, with a median of 4 clinical settings (interquartile range: 2–6). A total of 71 participants representing 49 different clinical settings completed the online survey: 37 participants from rehabilitation centres (16 managers; 21 clinicians) and 34 participants from health houses (11 managers; 23 clinicians). In total, 27 managers and 44 healthcare professionals provided survey data.

Background information on all participants is provided in Table 1. Most respondents were trained as nurses (64.8%; 46/71) and accounted for 77.8% (21/27) of all managers. Other respondent managers were trained physiotherapists (11.1%; 3/27), occupational therapists (7.4%; 2/27) and medical doctor (3.7%; 1/27). The respondent healthcare professionals consisted of nurses (56.8%; 25/44), speech therapists (20.4%; 9/44), physiotherapists (6.8%; 3/44), occupational therapists (6.8%; 3/44), nutritionists (6.8%; 3/44) and other allied health professionals (2.3%; 1/44). Completion of surveys took about fifteen minutes (median 15 min; interquartile range: 25–29).

### 3.2. Rehabilitation Centres and Health Houses

Rehabilitation centres and health houses offered different types of care. Almost all centres and health houses offered inpatient short-term stays (88.9%; 24/27) while about one third of all settings provided inpatient long-term stays (37.0%; 10/27) and day-care (33.3%; 9/27). Home healthcare and outpatient care were less common (respectively, 14.8% (4/27) and 11.1% (3/27)). The total number of beds per clinical setting varied greatly (Figure 1), ranging between 8 and 100 with a median of 29 beds (interquartile range: 17–62). The most frequent diagnostic groups admitted to rehabilitation centres and health houses were stroke (88.9%; 24/27) followed by neurodegenerative diseases (e.g., Parkinson’s disease, multiple sclerosis (85.2%; 23/27)) and traumatic brain injuries (85.2%; 23/27). Over 50% of all centres cared for oncology clients (55.6%; 15/27) and clients with congenital neurological conditions (e.g., cerebral palsy (55.6%; 15/27)) and less than 20% for people with dementia (18.5%; 5/27). Distribution of client numbers per diagnostic group are reported in Table 2.

Table 3 provides numbers and percentages of staff per rehabilitation centre and health house in full-time equivalents, including managerial positions, healthcare professionals, and personnel without a professional degree. All centres had managers, medical doctors, nurses and physiotherapists among their staff, and most centres also had occupational therapists (92.6%) present. Other, less frequently affiliated staff included: care assistants (66.7%), speech therapists (63.0%), nutritionists (44.4%), social workers (40.7%), personnel without a professional degree (33.3%), and psychologists (27%).

### 3.3. Screening and Assessment for Dysphagia

Managers (*n* = 27) were asked about how clients were screened or assessed for eating and swallowing difficulties in their clinical centres. Using a five-point ordinal scale (never, rarely, sometimes, often, always), they reported on the frequencies in which listed methods were used. Frequency percentages for the combined category ‘often-always’ varied between 41.7% and 69.6% for different methods and information sources: 69.6% (16/23) reports from external healthcare providers; 69.6% (16/23) conducted mealtime observation; 58.3% (14/24) screened for eating and swallowing difficulties; 54.5% (12/22) used client’s self-report or caregiver’s report; 52.4% (11/21) screened for malnutrition; and 41.7% (10/24) used a clinical assessment for eating and swallowing difficulties. Managers also provided data on whether screening and assessments were conducted routinely (‘often-always’) for specific impairments. These results indicated that 85.2% (23/27) screened for nutritional status; 76.9% (20/26) screened for the need to adjust consistency of food/drinks; 63.0% (17/27) screened for swallowing function; 63.0% (17/27) screened for the need to adjust medication intake (e.g., change of consistency, crushed tables); and 11.1% (11/27) screened for dental status. Different staff were involved in screening and assessment of eating and swallowing difficulties (‘often-always’), including speech therapists (67.2%; 33/64), nurses (51.6% 33/64), care assistants (23.5%; 12/51), nutritionists (19.5%; 8/41), occupational therapists (18.0%; 9/50), and personnel without a professional degree (5.0%; 2/40).

All participants (*n* = 71) commented on when clients were screened for eating and swallowing difficulties. About 20% (21.1%/ 15/71) of clients were screened before arrival to the rehabilitation centres and health houses, while about 50% of clients were screened immediately after arrival (50.7%; 36/71). Clients were also screened when staff observed changes in clients’ eating and swallowing behaviour (71.8%; 51/71) or changes in cognitive and/or physical functioning (19.7%; 14/71).

In addition, information was gathered from clinicians about decision-making processes on which clients to screen for eating and swallowing difficulties. In one out of five centres (20.0%; 8/40) all clients were screened routinely for problems with eating and drinking. In 50.0% (22/40) of health houses and rehabilitation centres, clients’ referrals included screening recommendations. Approximately 17.5% (7/40) of medical doctors reported they were responsible for the decision-making process and only very few healthcare professionals independently decided to screen clients (speech therapist and/or nurse (7.5%; 3/40)).

Many clients with eating and swallowing difficulties were assessed before arrival at the rehabilitation centres and health houses (56.8%; 25/44), while over 40% were assessed when arriving at the clinics (40.9%; 18/44). In addition, clients were assessed following observed changes in eating and drinking behaviours (70.4%; 31/44) or in cognitive or physical functioning (25.0%; 11/44).

Of all healthcare respondents, 40.0% (18/44) were directly involved in screening, listing non-standardised water swallows (40.4%; 8/18) as the most frequently used tool. Less frequently used screening tools included the Gugging Swallowing Screen (GUSS; 3/18), the Logemann four-finger method of palpation (2/18), the Toronto Bedside Swallowing Screening Test (TOR-BSST; 1/18), mealtime observation (1/18) and oral intake (2/18). Nine clinicians (12.7%; 9/44) were involved in assessment of eating and swallowing difficulties. The most frequently reported clinical assessment was the Mann Assessment of Swallowing Ability (MASA; 5/9), while the Radboud Oral Motor Inventory for Parkinson’s disease (ROMP; 2/9) and non-standardised clinical assessment including oral–motor examination (1/9) and mealtime observation (2/9) were reportedly used infrequently.

### 3.4. Dysphagia Management and Clinical Practice in Rehabilitation

Figure 2 and Table 4 present overviews of frequent (‘often-always’) challenges and difficulties experienced by clients with eating and swallowing problems as estimated by clinicians and managers. For all participants combined, most common challenges and difficulties were problems with medicine intake (56.1%; 37/66), oral residue (53.1%; 34/60), self-feeding or eye-hand coordination (51.7%; 31/60), reduced appetite (48.4%; 31/64), coughing during or after eating or drinking (47.0%; 31/66), and drooling (45.2%; 28/62). Other problems received following frequency scores: weight loss or malnutrition (41.5%; 27/65), communication problems (41.3%; 26/63), dehydration (31.2%; 20/64), chewing problems (20.7%; 12/58), changed or wet voice after drinking (18.9%; 10/53), and pneumonia (10.3%; 6/58).

Clinicians scored consistently higher frequencies compared to managers except for presence of pneumonia, dehydration, weight loss/malnutrition, and difficulties with self-feeding and eye-hand coordination, for which both respondent groups scored similar frequencies. Significant group differences were found for presence of drooling (two-tailed *p* < 0.001) and problems with medicine intake (two-tailed *p* = 0.020).

Table 5 presents an overview of strategies and routines used to support clients with eating and swallowing difficulties. The majority of clinicians (range: 50.0–97.6%; 20/40–41/42) reported to often or always use all but one (hand support during eating: 32.5%; 13/40) of the listed strategies and routines. However, when asking about treatment techniques in clients with eating and swallowing difficulties (Table 6), many clinicians failed to answer (range: 12.2–36.8%; 5/41–14/38). Estimated frequencies of clients being treated with various intervention techniques varied considerably. Among the respondents, the most commonly used techniques were oral motor exercises (44.7%; 17/38) and chin tuck (41.5%; 17/41); respondents reported that both techniques were used in the treatment of over 25% of all clients with eating and swallowing disorders. Neuromuscular electrical stimulation (NMES) was the least commonly used technique: 71.8% (28/39) of all respondents reported never using NMES. In addition, 36.8% (14/38) to 51.3% (20/39) of respondents reported they did not use swallowing manoeuvres/exercises (e.g., super supraglottic and supraglottic manoeuvre, Mendelsohn manoeuvre, Shaker exercise, effortful swallow, Masako manoeuvre), and about 40% reported not using head positioning, such as head tilt and head rotation, or thermal-tactile stimulation. Over 50% of clinical settings (54.5%; 24/44) have seldom or no access to external clinical professionals for assessment and treatment of clients with eating and swallowing difficulties, whereas about 20% reported to sometimes (22.7%; 10/44) or often to always (22.7%; 10/44) have access to external professionals.

Data were also retrieved on the mean treatment period for eating and swallowing difficulties, number of treatment sessions and session duration (Figure 3). Not all clinicians could provide the requested information (range: 22.7–38.6%; 10/44–17/44) and an additional seven respondents (15.9%; 7/44) indicated that their clients were not offered any treatment for eating and swallowing difficulties. Still, the most common treatment period reported was 2–4 weeks (27.3%; 12/44) or 5–8 weeks (20.5%; 9/44). The total number of sessions showed large variation but most frequently, clients received 2–5 treatment sessions (20.5%; 9/44) of half an hour (25.0%; 11/44). Clients were usually treated during individual sessions (always: 66.7%; 24/36) and occasionally in groups (never: 86.1%; 31/36).

Rehabilitation centres and health houses primarily employed kitchen staff to prepare meals for clients with eating and swallowing disorders (62.8%; 27/43). Over 25% of settings used external kitchen facilities (25.6%; 11/43), less than 10% employed staff other than kitchen staff (9.3%; 4/43), and in one centre clients’ relatives were involved (2.3%; 1/43). Settings used different resources when implementing recommendations for oral intake (eating and drinking). About 30% (29.5%; 13/44) reported using the International Dysphagia Diet Standardisation Initiative (IDDSI), about 20% referred to dietary handbooks (18.2%; 8/44), food and liquid classification systems developed by their own staff (20.4%; 9/44), or did not have any specific system in place (22.7%; 10/44).

### 3.5. Dysphagia Training and Education

Although few managers (14.8%; 4/27) and about one-third of all clinicians (29.5%; 13/44) reported required training and education, most centres (85.9%; 61/71) offered opportunities for professional development in eating and swallowing difficulties: theoretical upskilling (e.g., webinars, internal courses (67.6%; 48/71)), external expert course on eating and swallowing difficulties (40.8%; 29/71), coaching by experienced staff or colleagues (49.3%; 35/71), and onsite training (e.g., workshops (26.8%; 19/71)). Kitchen staff responsible for preparing meals for clients with eating and drinking difficulties received similar, but less frequent training and educational opportunities, namely: theoretical upskilling (31.0%; 22/71), external expert course on eating and swallowing difficulties (18.3%; 13/71), and onsite training (26.8%; 19/71). More than half of all respondents were unaware of any training for kitchen staff (52.1%; 37/71), and some respondents commented on not having kitchen staff employed (5.6%; 4/71).

### 3.6. Self-Perceived Quality of Dysphagia Care

At the end of both surveys, participants were asked how they perceived the quality of care for clients with eating and swallowing difficulties at their rehabilitation centre or health house using a five-point rating scale (incredibly good, good, neither bad nor good, poor, very poor). Almost half of all respondents perceived provided dysphagia care as incredibly good (9.9%; 7/71) or good (36.6%; 26/71). About 40% (39.4%; 28/71) rated care as neither bad nor good. Almost 15%, however, considered care to be either poor (11.3%; 8/71) or very poor (2.8%; 2/71). Group differences between managers and healthcare professionals were statistically significant (Chi-square test: χ^2^ [2, N = 71] = 7.216, *p* = 0.027), with managers perceiving quality of dysphagia care at a higher level compared to clinicians (Figure 4 and Figure 5). Group differences between health houses and rehabilitation centres were not significant (Chi-square test: χ^2^ [2, N = 71] = 2.963, *p* = 0.095).

## 4. Discussion

The main purpose of this study was to describe the management and care of dysphagia in rehabilitation centres and health houses across Norway. Two national, online surveys were developed, targeting either managers or healthcare professionals. The overall response rate was 72.1% and included 71 participants (27 managers and 44 healthcare professionals) from 49 different clinical sites. Participants working across all Norwegian counties provided data that are representative of Norwegian dysphagia care pathways in rehabilitation centres and health houses. Although substantial differences between healthcare settings were identified as well as between respondents (i.e., managers and clinicians), almost half of all respondents (46.5%) perceived provided dysphagia care as good or very good.

### 4.1. Staffing

As in many countries, speech therapists are the professionals most involved in dysphagia management and treatment in Norway. However, 37.0% of the rehabilitation centres and health houses did not have a speech therapist among their staff. Moreover, 54.5% of clinical settings reported seldom or never having access to any external clinical professionals for assessment and treatment of clients with eating and swallowing difficulties. Almost 16% of clinicians indicated that clients were not offered dysphagia treatment at all. These figures are concerning since the survey respondents indicated that the client populations most frequently admitted to rehabilitation centres and health houses had diagnoses of stroke (88.9%), neurodegenerative disease (85.2%) or traumatic brain injury (85.2%). In other words, based on reported diagnostic groups, prevalence of dysphagia in rehabilitation centres and health houses is expected to be high [7].

### 4.2. Screening and Assessment

Managers reported that clients in their rehabilitation centres and health houses were regularly screened and assessed for eating and swallowing difficulties (85.2%: often/always), of which 63.0% (always/often) claimed to screen for swallowing difficulties in particular. However, when asking clinicians, only 20.0% of respondents reported all clients were screened for eating and swallowing difficulties. While the most commonly used screening tools were non-standardised water swallows (40.4%) with unknown diagnostic performance, only limited data on implemented assessments were provided, with most clinicians indicating they were not involved in assessing clients. As respondents had specifically been recruited with knowledge about management and care of clients with dysphagia, this lack of information was unexpected and may indicate a restricted assessment of swallowing function after having failed screening. Moreover, even though the surveys contained short explanations on topics and terminology, such as ‘screening’ and ‘assessment’, clinicians seemed confused about differentiating between both terms, as screening tools were listed as assessments, and vice versa. Furthermore, none of the rehabilitation centres or health houses reported having access to either videofluoroscopy or endoscopy of swallowing, the so-called ‘gold standard’ assessments in dysphagia [23]. These findings are in line with The Norwegian National Guidelines for Treatment and Rehabilitation of Stroke [24], stating that only larger or specialised hospitals in Norway may have access to videofluoroscopy. This is quite disconcerting, as instrumental assessment can diagnose aspiration (including silent aspiration) and other physiological problems in the pharyngeal phase which cannot be confirmed by non-instrumental dysphagia assessment alone [23].

### 4.3. Intervention

Clinicians and managers were asked to estimate frequencies of challenges and difficulties experienced by clients with eating and swallowing problems. Clinicians scored consistently higher frequencies compared to managers except for challenges associated with the presence of pneumonia, dehydration, weight loss/malnutrition, and difficulties with self-feeding and eye-hand coordination. For these challenges and difficulties, both respondent groups scored very similar frequencies. Possibly, clinicians report higher frequencies as they are more familiar with dysphagia-related problems and trained in recognising challenges and difficulties resulting from eating and swallowing problems.

When asking clinicians about strategies and routines to support clients with eating and swallowing difficulties, most respondents reported high frequent use of almost all techniques listed. For example, over 95% of respondents (95.3%) reported to often or always thicken liquids, despite an increasing number of studies suggesting that the use of texture-modified liquids lacks sufficient scientific evidence for reducing pneumonia in clients with dysphagia. Thickened liquids, however, may result in reduced fluid intake, undernutrition, and result in decreased clients’ health-related quality of life [25,26,27,28]. Oral care after meals was comparatively uncommon according to respondents, with only 50% of respondents reporting they would often or always offer oral care after meals while the complementary 50% would only sometimes (15.0%), rarely (20.0%) or never (10.0%) support oral care. As poor oral hygiene is associated with increased risk of aspiration pneumonia [29], these data are disturbing. When asking about specific interventions, many clinicians (ranging between 12.2% and 36.8%) were unaware of which other treatment techniques were used. Of those clinicians providing additional intervention data, over 40% reported oral motor exercises (44.7%) and chin tuck (41.5%) to be the most used interventions in over 25% of all clients with eating and swallowing difficulties. Swallow manoeuvres, on the other hand, were less commonly used, and in 36.8% to 51.3% percent of clients swallow manoeuvres were never applied. The rare use of electrical stimulation (71.8%: never) was in line with previous studies (e.g., [15]).

Based on the current survey data, clients with dysphagia seem to be mainly supported using compensatory strategies and routines, including bolus modification. Further, the use of rehabilitative interventions, such as swallow manoeuvres, seem to very infrequently be part of dysphagia treatment in rehabilitation centres or health houses.

### 4.4. Education

Most rehabilitation centres and health houses (85.9%) provided opportunities for professional development in eating and swallowing difficulties for staff. The most common routines for upskilling staff were theoretical upskilling, such as webinars and internal courses (76.6%), coaching by experience staff or colleagues (49.3%), and external expert courses on eating and swallowing difficulties. Kitchen staff (i.e., those responsible for preparing meals for clients with dysphagia) were given fewer training and educational opportunities (most frequently theoretical upskilling 31.0% and onsite training 26.8%). However, over 50% of all respondents were unaware if kitchen staff received any training. Additionally, 22.7% of respondents indicated not having any specific food and liquid classification system in place, thus contributing to challenges in standardising the preparation of meals for clients with dysphagia.

### 4.5. Quality of Care

Managers perceived the quality of dysphagia care at a significantly higher level compared to clinicians. However, managers also scored consistently lower frequencies for most challenges and difficulties as experienced by clients with dysphagia. Possibly, awareness of dysphagia and its symptoms and consequences, may have influenced the quality of care ratings.

### 4.6. Limitations

Response-rate-induced bias is not necessarily an important threat to the validity of questionnaires, but may affect whether the responses represent people’s true states, attitudes, and behaviours [30]. The current study achieved an overall response rate of 72.1%, representing 55 out of 68 identified rehabilitation centres and health houses. The respondents are considered to be a representative selection of targeted healthcare settings across Norway and reflect a good response rate when compared to similar survey studies [9,10].

Furthermore, when developing both surveys, a compromise was struck between respondent burden and data retrieval. Adding more questions would have contributed to the required time for survey completion and have negatively affected the response rate. However, future studies may consider including more respondents per healthcare setting, aiming to differentiate dysphagia management and care as provided and perceived by different professional groups.

## 5. Conclusions

Significant differences in dysphagia care and management were identified across rehabilitation in Norway. Rehabilitation centres and health houses often did not have a speech therapist among their staff or access to external clinical professionals. In addition, over fifteen percent of clinicians reported that no dysphagia treatment was offered even though client populations included in rehabilitation suggested high dysphagia prevalence.

In most centres, screening was conducted using non-standardised water swallows with unknown diagnostic performance. None of the respondents reported having access to instrumental assessment and only limited information was available on non-instrumental assessment processes. Dysphagia interventions mainly consisted of compensatory strategies and routines, including bolus modification, with very infrequent use of rehabilitative interventions, such as swallow manoeuvres.

Although almost half of all respondents perceived the overall quality of care for clients with eating and swallowing problems as good, lack of awareness of dysphagia and its symptoms, consequences, and options for treatment, may have influenced quality ratings. There is a need to raise awareness of dysphagia and provide training opportunities for healthcare professionals in both screening and assessment, and dysphagia care and management.

## Figures and Tables

**Figure 1 jcm-11-05730-f001:**
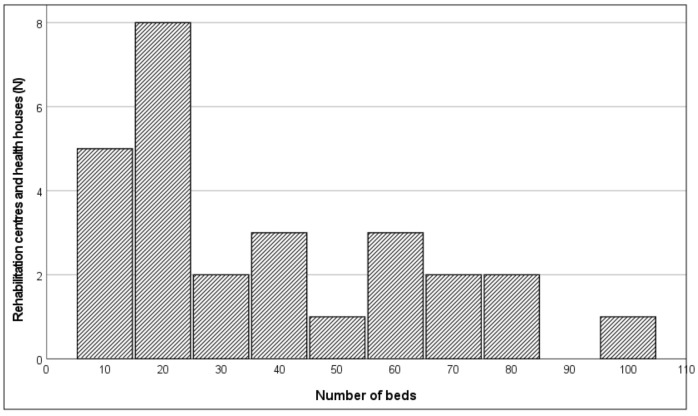
Number of beds per rehabilitation centre and health house (*n* = 27).

**Figure 2 jcm-11-05730-f002:**
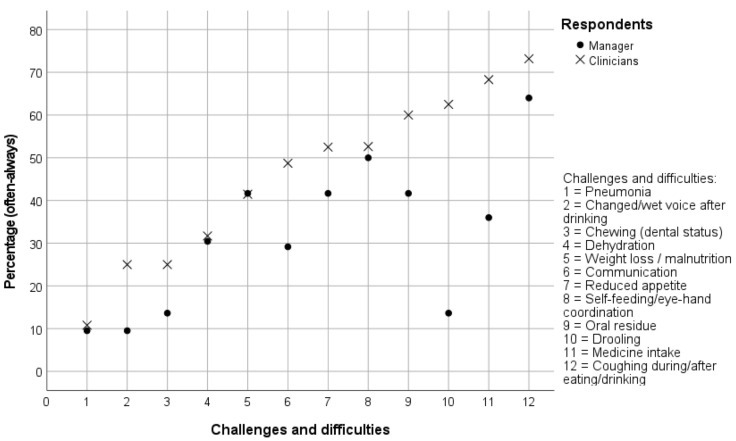
Frequency percentages (score ‘often-always’) of managers and clinicians on challenges and difficulties for people with eating and swallowing problems.

**Figure 3 jcm-11-05730-f003:**
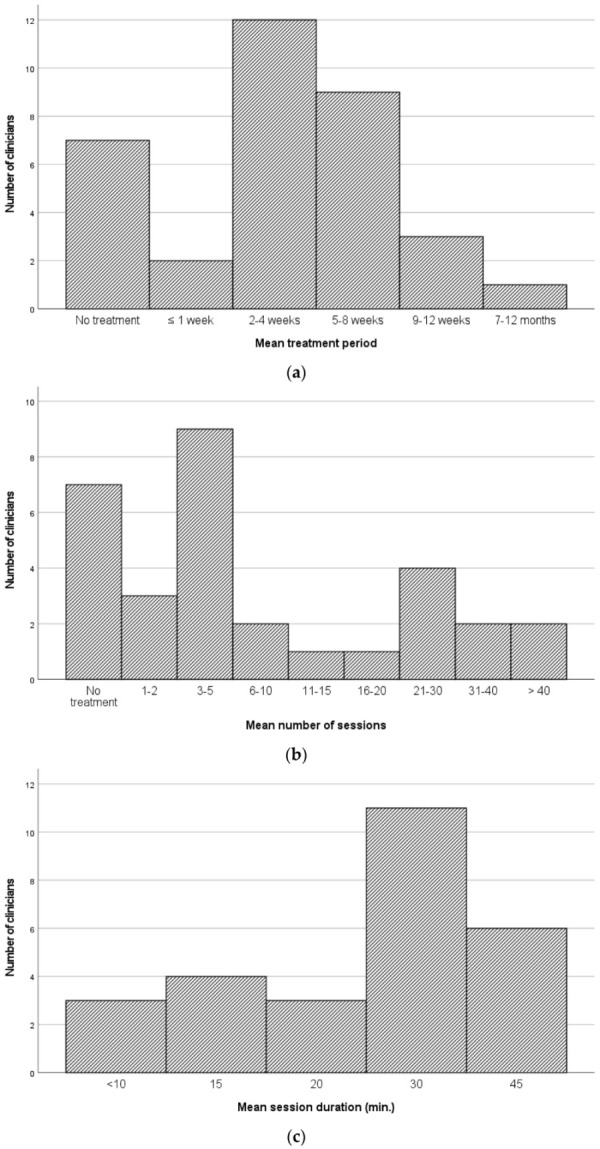
(**a**) Mean treatment period for clients with eating and swallowing difficulties (*n*_Clinicians_ = 34). (**b**) Mean number of sessions for clients with eating and swallowing difficulties (*n*_Clinicians_ = 41). (**c**) Mean sessions duration for clients with eating and swallowing difficulties (*n*_Clinicians_ = 27).

**Figure 4 jcm-11-05730-f004:**
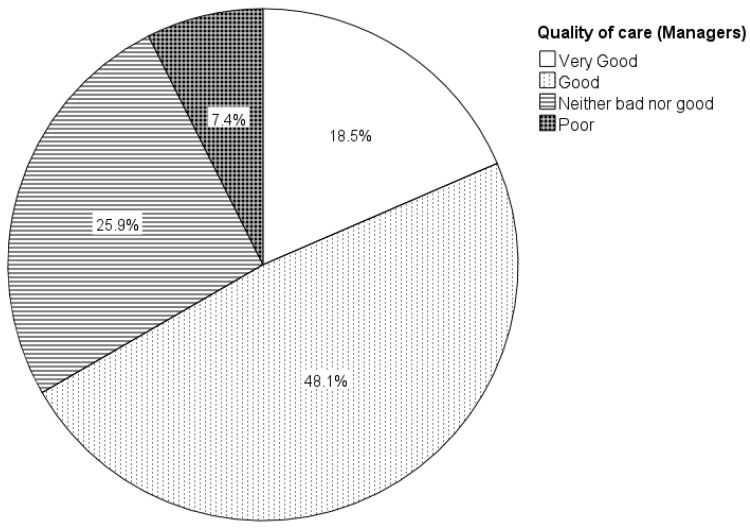
Managers’ (*n* = 27) self-perceived quality of care for people with eating and swallowing difficulties in rehabilitation centres and health houses. *Note*. Numbers are rounded to one decimal place.

**Figure 5 jcm-11-05730-f005:**
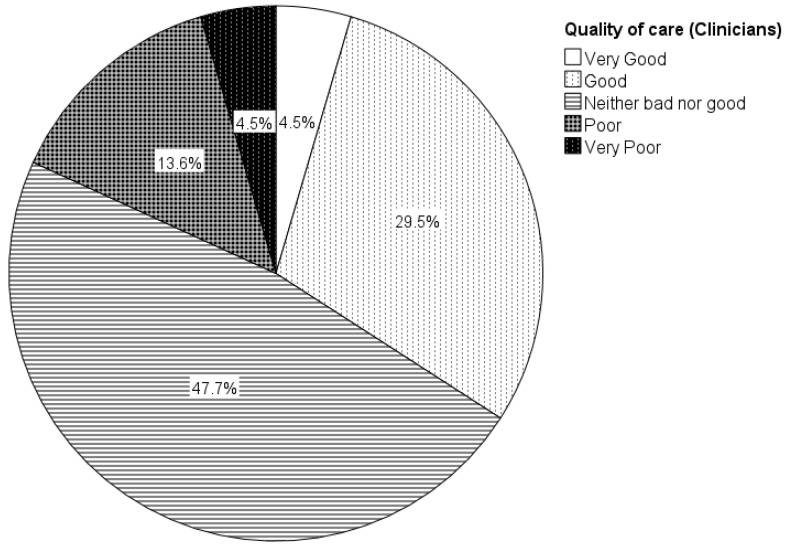
Clinicians’ (*n* = 44) self-perceived quality of care for people with eating and swallowing difficulties in rehabilitation centres and health houses. *Note*. Numbers are rounded to one decimal place.

**Table 1 jcm-11-05730-t001:** Position and professional background of respondents (*n* = 71).

Position	Educational Background							Total Number of Respondents (%)
	Medical Doctor	Nurse	Physio- therapist	Occupational Therapist	Speech Therapist	Nutritionist	Other Allied Health	
Executive officer	0	0	0	0	0	0	1	**1 (1.4%)**
Department manager	** *1 ** **	** *17 ** **	** *2 ** **	** *2 ** **	0	0	0	**22 (31.0%)**
Head of allied health	0	** *1 ** **	** *1 ** ** **+ 1**	0	0	1	0	**4 (5.6%)**
Head of nursing	0	** *2 ** **	0	0	0	0	0	**2 (2.8%)**
Nurse	0	***1 **** + 25	0	0	0	0	0	**26 (36.6%)**
Speech therapist	0	0	0	0	9	0	0	**9 (12.7%)**
Physiotherapist	0	0	2	0	0	0	0	**2 (2.8%)**
Occupational therapist	0	0	0	3	0	0	0	**3 (4.2%)**
Nutritionist	0	0	0	0	0	2	0	**2 (2.8%)**
**Ratio managers/clinicians**	**1 */0**	**21 */25**	**3 */3**	**2 */3**	**0 */9**	**0 */3**	**0 */1**	**27 */44 (38%/62%)**
**Total number of respondents (%)**	**1 (1.4%)**	**46 (64.8%)**	**6 (8.4%)**	**5 (7.0%)**	**9 (12.7%)**	**3 (4.2%)**	**1 (1.4%)**	**71 (100%)**

* Participants mainly involved in managerial activities.

**Table 2 jcm-11-05730-t002:** Diagnostic groups in rehabilitation centres and health houses.

Diagnostic Group	Number (%)										Number of Respondents (%)
	0 (%)	1–5 (%)	6–10 (%)	11–20 (%)	21–30 (%)	31–40 (%)	41–60 (%)	61–80 (%)	≥81 (%)	Unknown	
Dementia	13 (48.2%)	4 (14.8%)	0	1 (3.7%)	5 (18.5%)	2 (7.4%)	0	0	1 (3.7%)	1 (3.7%)	26/27 (96.3%)
Neurodegenerative diseases	4 (14.8%)	3 (11.1%)	4 (14.8%)	7 (25.9%)	2 (7.4%)	3 (11.1%)	0	0	1 (3.7%)	3 (11.1%)	24/27 (88.9%)
Traumatic brain injury	3 (11.1%)	9 (33.3%)	8 (29.6%)	3 (11.1%)	0	1 (3.7%)	0	0	0	3 (11.1%)	24/27 (88.9%)
Stroke	3 (11.1%)	0	3 (11.1%)	5 (18.5%)	5 (18.5%)	4 (14.8%)	3 (11.1%)	1 (3.7%)	1 (3.7%)	2 (7.4%)	25/27 (92.6%)
Oncology	11 40.7%)	9 (33.3%)	1 (3.7%)	3 (11.1%)	1 (3.7%)	0	1 (3.7%)	0	0	1 (3.7%)	26/27 (96.3%)
Congenital neurological conditions	13 (48.2%)	7 (25.9%)	3 (11.1%)	0	1 (3.7%)	0	0	0	0	3 (11.1%)	23/27 (85.2%)

**Table 3 jcm-11-05730-t003:** Staffing in rehabilitation centres and health houses (*n*_respondents_ = 27).

Staff	Number of Staff in FTE * (%)							
	0 (%)	1 (%)	2–5 (%)	6–10 (%)	11–15 (%)	16–25 (%)	26–50 (%)	≥51 (%)
Manager	0	13 (48.2%)	11 40.7%)	2 (7.4%)	0	0	1 (3.7%)	0
Medical doctor	0	10 (37.0%)	17 (63.0%)	0	0	0	0	0
Nurse	0	0	4 (14.8%)	6 (22.2%)	5 (18.5%)	10 (37.0%)	2 (7.4%)	0
Speech therapist	10 (37.0%)	12 (44.4%)	5 (18.5%)	0	0	0	0	0
Physiotherapist	0	6 (22.2%)	13 (48.2%)	2 (7.4%)	4 (14.8%)	2 (7.4%)	0	0
Occupational therapist	2 (7.4%)	6 (22.2%)	17 (63.0%)	1 (3.7%)	0	0	0	0
Nutritionist	15 (55.6%)	10 (37.0%)	2 (7.4%)	0	0	0	0	0
Social worker	16 (59.3%)	6 22.2%)	5 (18.5%)	0	0	0	0	0
Psychologist	17 (63.0%)	7 (25.9%)	3 (11.1%)	0	0	0	0	0
Care assistants	9 (33.3%)	1 (3.7%)	11 (40.7%)	3 (11.1%)	2 (7.4%)	0	1 (3.7%)	0
Personnel without a professional degree	18 (66.7%)	1 (3.7%)	5 (18.5%)	1 (3.7%)	2 (7.4%)	0	0	0

* Full-time equivalent.

**Table 4 jcm-11-05730-t004:** Frequencies (‘Often-always’) of challenges and difficulties for people with eating and swallowing problems.

Challenges/Difficulties	Frequency ‘Often-Always’: *n* (%)			Fisher’s Exact Test
	Clinicians (*n* = 44)	Managers (*n* = 27)	All (*n* = 71)	*p*-Value (2-Sided)
Pneumonia	10.8% (4/37)	9.5% (2/21)	10.3% (6/58)	ns
Changed/wet voice after drinking	25.0% (8/32)	9.5% (2/21)	18.9% (10/53)	ns
Chewing problems (caused by dental status)	25.0% (9/36)	13.6% (3/22)	20.7% (12/58)	ns
Dehydration	31.7% (13/41)	30.4% (7/23)	31.2% (20/64)	ns
Weight loss or malnutrition	41.5% (17/41)	41.7% (10/24)	41.5% (27/65)	ns
Communication problems	48.7% (19/39)	29.2% (7/24)	41.3% (26/63)	ns
Reduced appetite	52.5% (21/40)	41.7% (10/24)	48.4% (31/64)	ns
Problems self-feeding or eye-hand coordination	52.6% (20/38)	50.0% (11/22)	51.7 (31/60)	ns
Food residues in mouth after swallowing	60.0% (24/40)	41.7% (10/24)	53.1% (34/64)	ns
Drooling	62.5% (25/40)	13.6% (3/22)	45.2% (28/62)	<0.001 *
Coughing during/after eating or drinking	73.2% (30/41)	64.0% (16/25)	47.0% (31/66)	ns
Problems medicine intake	68.3% (28/41)	36.0% (9/25)	56.1% (37/66)	0.020 *

*Note*. ns = not significant. * significant (*p* ≤ 0.050).

**Table 5 jcm-11-05730-t005:** Strategies and routines to support clients with eating and swallowing difficulties (*n*_Clinicians_ = 44).

Strategy/Routine	Frequency % (*n*)						
	Never	Rarely	Sometimes	Often	Always	Unknown	*n* _Total_
Improving clients’ upright sitting posture	0	0	2.4% (1)	28.6% (12)	** *69.0% (29)* **	0	42
Adjusting head positioning	0	5.0% (2)	10.0% (4)	25.0% (10)	55.0% (22)	5.0% (2)	40
Use of customized mealtime utensils	0	4.7% (2)	30.2% (13)	** *46.5% (20)* **	18.6% (8)	0	43
Modification of food consistencies	2.3% (1)	2.3% (1)	0	** *65.1% (28)* **	30.2% (13)	0	43
Modification of liquid consistencies	0	2.3% (1)	9.3% (4)	** *62.6% (27)* **	25.6% (11)	0	43
Change of medicine intake (e.g., change of consistence, crushed tablets, or liquid instead of tablets)	4.7% (2)	4.7% (2)	18.6% (8)	** *53.5% (23)* **	16.3% (7)	2.3% (1)	43
Changes in the environment (avoidance of distracting background activities or noise, e.g., television, music)	0	4.9% (2)	34.1% (14)	** *39.0% (16)* **	19.5% (8)	2.4% (1)	41
Mealtime observation	2.4% (1)	2.4% (1)	14.3% (6)	** *40.5% (17)* **	** *40.5% (17)* **	0	42
Checking for clients to be well rested and alert during mealtimes	0	9.5% (4)	16.7% (7)	** *40.5% (17)* **	28.6% (12)	4.8% (2)	42
Offering hand support during eating	0	25.0% (10)	** *40.0% (16)* **	27.5% (11)	5.0% (2)	2.5% (1)	40
Control of bolus size per mouthful	5.0% (2)	12.5% (5)	15.0% (6)	** *42.5% (17)* **	25.0% (10)	0	40
Checking for food residues in mouth	0	22.2% (9)	15.0% (6)	20.0% (8)	** *40.0% (16)* **	2.5% (1)	40
Controlling speed of oral intake	0	7.5% (3)	27.5% (11)	** *32.5% (13)* **	27.5% (11)	5.0% (2)	40
Having clients actively engaged in drinking and eating activities	0	5.0% (2)	15.0% (6)	** *47.5% (19)* **	27.5% (11)	5.0% (2)	40
Allowing prolonged upright sitting after mealtimes for at least 15 min	0	7.3% (3)	26.8% (11)	29.3% (12)	** *31.7% (13)* **	4.9% (2)	41
Oral care after meals	10.0% (4)	20.0% (8)	15.0% (6)	** *40.0% (16* ** *)*	10.0% (4)	5.0% (2)	40

*Note*. Highest frequencies per strategy/routine in bold-italics.

**Table 6 jcm-11-05730-t006:** Treatment techniques in clients with eating and swallowing difficulties (*n*_Clinicians_ = 44).

Treatment Technique	Frequency % (*n*)							
	0%	1–10%	11–25%	26–50%	51–75%	>75%	Unknown	*n* _Total_
Oral motor exercises	13.2% (5)	** *23.7% (9)* **	5.3% (2)	18.4% (7)	7.9% (3)	18.4% (7)	13.2% (5)	38
Super supraglottic manoeuvre and supraglottic manoeuvre	** *36.8% (14)* **	10.5% (4)	2.6% (1)	7.9% (3)	5.3% (2)	2.6% (1)	34.2% (13)	38
Mendelsohn manoeuvre	** *43.6% (17)* **	5.1% (2)	5.1% (2)	7.7% (3)	7.7% (3)	2.6% (1)	28.2% (11)	39
Shaker exercise	** *48.7% (19)* **	5.1% (2)	7.7% (3)	5.1% (2)	0	0	33.3% (13)	39
Effortful swallow	** *38.5% (15)* **	0	5.1% (2)	10.3% (4)	0	12.8% (5)	33.33% (13)	39
Masako manoeuvre	** *51.3% (20)* **	0	2.6% (1)	2.6% (1)	10.3% (4)	0	33.3% (13)	39
Other swallowing manoeuvres	** *45.7% (16)* **	8.6% (3)	5.7% (2)	5.7% (2)	0	0	34.3% (12)	35
Chin tuck	19.5% (8)	14.6% (6)	12.2% (5)	** *24.4% (10)* **	4.9% (2)	12.2% (5)	12.2% (5)	41
Head tilt	** *40.0% (16)* **	20.0% (8)	5.0% (2)	15.0% (6)	0	2.5% (1)	17.5% (7)	40
Head rotation	** *41.0% (16)* **	15.4% (6)	7.7% (3)	10.3% (4)	0	2.6% (1)	23.1% (9)	39
Other changes in head positioning	** *42.5% (17)* **	15.0% (6)	7.5% (3)	12.5% (5)	0	2.5% (1)	20.0% (8)	40
Thermal-tactile stimulation	** *39.5% (15)* **	5.3% (2)	5.3% (2)	2.6% (1)	5.3% (2)	5.3% (2)	36.8% (14)	38
Neuromuscular electrical stimulation (NMES)	** *71.8% (28)* **	2.6% (1)	2.6% (1)	0	0	0	23.1% (9	39

*Note*. Highest frequencies per treatment technique in bold-italics.

## Data Availability

The data presented in this study are available on request from the first author.

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
