# Peer review of "Dysphagia Care and Management in Rehabilitation: A National Survey"

_jcm, 2022, doi:10.3390/jcm11195730_

Round 1

Reviewer 1 Report

Clearly reported study design and findings, comments for the authors' consideration:

Recruitment of participants - Please add here that when recruiting, the researchers asked for survey responders with knowledge of dysphagia management. This was stated in discussion but not methods.

Results

- the use of the term "unskilled personnel" seems dismissive. Does the "unskilled" part relate specifically to dysphagia management and the personnel work in supportive/admin roles in the facilities' day to day running? Perhaps this is a Norwegian to English translation issue. I think it's important to acknowledge that all roles have skills attached even if they don't come with a professional degree.

Page 6, last paragraph - the use of the fraction 1-3/18 - can you specify the individual fractions for each variable to allow readers to see how they add up instead of lumping them together like this? Same for 1-2/9 in the same paragraph continued on page 7.

Page 9 - "Not all clinicians could provide the requested information (range: 22.7% - 38.6%; 10/44 – 17/44) and an additional seven respondents (15.9%; 7/44) who indicated that THEIR clients were not offered any treatment for eating and swallowing difficulties."

Page 9 last paragraph - suggest to rewrite the sentence "Over 25% used external kitchen facilities (25.6%; 11/43), staff other than kitchen staff (9.3%; 4/43) or clients’ relatives (2.3%; 1/43)."

Reviewer 2 Report

Abstract.
clients is not suitable term. Better use patients

Introduction.

the classification of the severity of dysphagia needs to be developped.

also, the psychological dysphasia is frequent. Better to mention it and was it included in the survery ? Were the patients excluded ?

Material and methods.

The materials and methods section is despicable. However, it is not showing the content of the survery.

can the survey be added as a figure or table ? 

discussion
the layout of the article must be corrected: 1. Introduction, 2. Material and methods ..

discussion should not be written in paragraphs.

hence, overall the paper is very well written and very interesting. The content is very scientific and of value. 
